# Psychometric Properties of the Turkish Version of the Rating Scale of Pain Expression during Childbirth Scale (ESVADOPA-TR)

**DOI:** 10.3390/healthcare12171745

**Published:** 2024-09-02

**Authors:** Burcu Avcıbay-Vurgeç, Silvia Navarro-Prado, Sule Gökyıldız-Sürücü, Muhsin Dursun, María Angustias Sánchez-Ojeda

**Affiliations:** 1Department of Midwifery, Faculty of Health Science, Cukurova University, 01330 Adana, Turkey; burcuavcibay@hotmail.com (B.A.-V.); gokyildizsule@gmail.com (S.G.-S.); 2Department of Nursing, Faculty of Health Sciences, Melilla Campus, University of Granada, 52005 Melilla, Spain; maso@ugr.es; 3Department of Orthopedics and Traumatology, Adana Ortadogu Special Hospital, 67055 Adana, Turkey; muhsindursun0@yahoo.com

**Keywords:** labor, pain, psychometric properties, validation, reliability

## Abstract

Background: Evaluation of a specific and dynamic pain, such as labor pain, with a situation-specific measurement tool will increase the quality of care given during childbirth. The Rating Scale of Pain Expression during Childbirth (ESVADOPA) is a situation-specific measurement tool for labor pain. The aim was to examine the psychometric properties of a Turkish version of the ESVADOPA scale. Methods: This study utilized a methodological design. Participants were 158 pregnant women at term and in spontaneous labor. Two measurements were performed during the passive and active phases of labor. To avoid bias between the raters, all the evaluations were performed by a single midwife. Validity analyses of the scale were performed using exploratory factor analysis and confirmatory factor analysis. Results: The scale was composed of a one-factor structure that had an eigenvalue of over 1 and explained 71.79% of the variance. Cronbach’s alpha internal consistency coefficient of the scale is 0.92. An analysis of the factor structure showed that the item factor loads ranged between 0.729 and 0.897. In the confirmatory factor analysis, the results showed that the data had a good fit with the model. Conclusions: The Turkish version of the ESVADOPA was found to have high reliability and validity for determining the expression of pain during childbirth.

## 1. Introduction

Pain is a common symptom reported by pregnant women as part of the childbirth experience. Childbirth pain has consistently been among the most severe pain types to be experienced by a woman throughout her life [1,2]. This pain, which is part of the childbirth process, is an acute pain that is neither dangerous nor threatening. It is unique as it is perceived as necessary, and compared to other pain types, it is more easily accepted. It enables experience-based information about the normal function of the process. It is a complicated, subjective, multidimensional response to sensory stimuli during childbirth [3]. In addition to the current childbirth experience, many previous cultural, social, emotional, sensory, physical, and biochemical factors, singly or in combination, could affect the individual pain experienced. For this reason, like in all pain types, one of the challenging characteristics of childbirth pain is its subjective nature; namely, it cannot directly be observed by those who have not experienced it [4,5]. Childbirth pain could affect not only the mother’s physical health, her emotional relationship with the child, and thoughts about childbirth in the future but also her childbirth satisfaction and comfort. For this reason, childbirth pain management is important in terms of both medical and moral aspects.

The most commonly used tools for measuring childbirth pain include standardized tools such as the Visual Analog Scale (VAS), Numerical Rating Scale (NRS), Verbal Rating Scale (VRS), or Non-Verbal Pain Scale (NVPS). The dynamic and progressive nature of childbirth pain is insufficiently captured by these kinds of pain severity scales [6]. Moreover, they do not consider interactions among factors such as contraction severity, the presence of any relaxation between the contractions, the relative duration of this relaxation, total childbirth pain duration and coping, intrapartum support, and sleep deprivation [7]. The McGill Pain Questionnaire (MPQ) was developed to improve the analog pain severity scales [8]. In addition to the MPQ pain severity, a total of 78 adjectives were defined in 20 categories, including the verbal pain descriptors and pain-related emotional components. However, most items of this scale that can be utilized for assessing childbirth pain are slightly associated with the woman’s childbirth pain experience. The childbirth pain experience is specific, multidimensional, and complicated [7]. Therefore, there is a significant need in the literature for a specific tool to measure labor pain. Despite the continuous increase in practices and options for eliminating childbirth pain in modern and traditional practices, studies that specifically focused on suitable tools for rating labor pain have existed only within the last decade. The Angle Labor Pain Questionnaire (A-LPQ), the first measurement tool to measure labor pain, was developed by Angle et al. The A-LPQ was developed both to assess labor pain and to define non-progressive labor caused by fetal malposition during the delivery process. Similar to other measurement tools, it is a measurement tool based on the pregnant woman’s evaluation [9,10]. However, the laboring woman’s evaluation capacity may change as the labor progresses, and there could be cases in which these kinds of scales are suitable for only the initiation of labor. Conditions may not be convenient for the woman to respond to the measurement questions. Therefore, they can be insufficient in evaluating the pain occurring close to the expulsion phase. Moreover, the laboring woman may not speak the same language as the health professionals providing care. The Rating Scale of Pain Expression during Childbirth (in Spanish, Escala de Valoración de la Expresión del Dolor durante el Trabajo de Parto—ESVADOPA) is a multidimensional scale specific to childbirth pain and based on the assessment of the caregiver [1]. ESVADOPA is completed merely based on the midwife’s observation; it is a cost-effective and rapidly applied scale developed for the expression of childbirth pain that is administered without intervening in the childbirth process or being affected by a language barrier.

The purpose of this study is to perform psychometric analyses and test the reliability and validity of the Turkish version of the “Rating Scale of Pain Expression during Childbirth”, which was developed specifically to evaluate childbirth pain and is currently being adapted to Chinese, Brazilian, Portuguese, and Italian languages.

## 2. Materials and Methods

### 2.1. Design

This study utilized a methodological design to evaluate the intercultural adaptation, reliability, and validity of the ESVADOPA scale, which will be used by midwives while evaluating childbirth pain.

### 2.2. Participants

This study was conducted in the delivery room of the Obstetrics and Gynecology hospital between 1 January and 31 June 2021. The target population of this study was pregnant women in the childbirth process, and the sample included pregnant women who came to the hospital for childbirth and were randomly selected between the dates when this study was conducted. The sample size recommended in the literature for methodological studies was utilized for calculating the sample size of this study. ESVADOPA evaluates the expression of pain during childbirth in six items. These items include facial muscles, body response, verbal response, restlessness, the ability to relax, and vegetative symptoms. Each item includes four sub-items that make assessments at different levels. According to the intraclass correlation coefficient (ICC), the acceptable ICC = 0.70, ICC = 0.70, expected ICC = 0.80, significance level = 0.05, and 80% power, the sample size for the 6-item scale was calculated as 74. The sample size was doubled for further analysis [11]. In light of this information, considering the potential data loss, the sample size was determined to be 150 participants, and this study was completed with 158 participants. The city where this study was conducted was densely populated with Syrian immigrants. To emphasize the feature of the scale eliminating the language barrier, this study also included foreign pregnant women who did not speak Turkish. The inclusion criteria of this study were as follows:Primigravida;38–40 weeks of gestation;Spontaneous labor;Cervical dilatation between 2 and 8 cm;Absence of epidural anesthesia or any intervention to relieve labor pain.

### 2.3. Translation Process

Written approval was obtained from the authors of the original scale to use the scale, and the present study was co-authored by the authors of the original scale. The first phase of this study was to translate and adapt the scale to use in Turkish. The scale items were initially translated into Turkish by the researchers. Then the scale was translated from English to Turkish by three instructors in the Department of Midwifery who were specialized in the field and competent in two languages (English and Turkish). After all the translated versions were revised by the researchers, the most suitable statements were chosen to form the Turkish version of the scale. This form was assessed by a Turkish language expert, and necessary revisions were made based on her suggestions. Back translation of the scale from Turkish to English was performed by two experts who had not seen the original scale before and were competent in two languages (English and Turkish). After the translation procedure, the scale was administered to 10 pregnant women who met the sample characteristics and were not included in this study by 10 different midwives to pilot test its comprehensibility. Expert views were received for evaluating the content validity of the scale. To accomplish this, the Turkish version of the scale after the translation process was submitted to the expert views of 9 academic researchers in the fields of midwifery (2), gynecological nursing (2), gynecology (2), physiotherapy (1), anatomy (1), and psychology (1) for the preparation of its final form.

### 2.4. Data Collection and Tools

Data collection was carried out during the first stage of labor. To test the invariance of the scale over time, two measurements were performed at passive and active labor phases. The passive phase was the time from the beginning of labor to 5 cm cervical dilatation, while the active phase began at 6 cm cervical dilatation to birth. To avoid bias between the raters, all the evaluations were carried out by a single midwife. This study utilized the Participant Information Form, Visual Analog Scale (VAS), and Rating Scale of Pain Expression during Childbirth (ESVADOPA).

The Patient Information Form was composed of 7 items. The form includes questions concerning the obstetric and demographic characteristics of pregnant women.

VAS is a commonly used scale in the field of obstetrics to measure pain. It is a 10 cm [100 mm] ruler, indicating no pain on one end and the worst pain possible on the other end. The patient determines the pain she experiences by marking a line, point, or any signs on the ruler. The distance from no pain to the place marked by the patient is measured in centimeters, and the numerical data found show the severity of the patient’s pain. VAS was first developed by Price et al. [12]. It is a reliable measurement tool that is commonly used to assess subjective pain. In this study, it was used as a convergent assessment tool.

ESVADOPA, developed by Navarro-Prado et al., is a scale that is used without intervening in the labor process or affected by a language barrier and is completed only based on the midwife’s observation to rate the expression of pain during labor [1]. The scale includes 6 items to be evaluated, which include facial muscles, body response, verbal response, restlessness, ability to relax, and vegetative symptoms. Each item includes four sub-items scored between 0 and 3. When the sub-items are scored, “0” indicates lack of pain and “3” indicates maximum pain. There is no need to establish verbal communication with the women to fill in the scale. The assessment requires observing the woman’s response only during the contraction. If any one of the items is unclear, then the evaluation is performed based on successive contractions. The scale total score is obtained when the scores are summed after all the items are assessed. The pain level is determined according to the categorized scoring, which includes <1: does not express pain; 1–6: expresses mild pain; 7–12: expresses moderate pain, and 13–18: expresses intense pain. The evaluation is performed in the active phase of labor with the absence of a desire to push. The scale should be administered without the presence of epidural anesthesia or any other obstetric interventions.

### 2.5. Data Analysis

Data were analyzed in SPSS 25 and AMOS 25 programs. Data for descriptive characteristics were presented using frequencies and percentages. A Shapiro–Wilks test was performed to determine the normality distribution. The Shapiro–Wilks test was performed to determine the normality distribution. Since the data showed a normal distribution, parametric tests were applied in the analysis of the data. A Kaiser–Meyer–Olkin (KMO) test was performed to determine if the sample size in reliability and validity analyses was adequate for EFA. The Davis technique was utilized for the analysis of the content validity index (CVI). Bartlett’s Test of Sphericity was used to test if there was a correlation between the variables. Validity analyses of the scale were performed using exploratory factor analysis (EFA) and confirmatory factor analysis (CFA). Convergent or concurrent validity was performed using the Pearson correlation coefficient. Two measurements during the passive and active phases were carried out for test–retest. The maximum likelihood test was used for test–retest reliability measurements. Statistical significance was accepted at *p* < 0.05.

### 2.6. Ethical Considerations

Permission was obtained for the Turkish version by contacting the authors of the scale. This study was co-authored with the researchers. Ethical approval for this study was obtained from the Cukurova University Medical Faculty Non-Invasive Clinical Research Ethics Committee (approval number: 04 December 2020-106/25). Before applying the questionnaires, all participants were informed about the purpose of this study, and written informed consent was signed for participation and dissemination of the results.

## 3. Results

This study was conducted with 158 pregnant women aged between 18 and 44 (mean = 26.87 ± 6.50) whose labor process had begun spontaneously. Of all the participants, 98 were Turkish citizens, and the rest of the patients were Syrian pregnant women who did not speak fluently Turkish or spoke only Arabic (Table 1). Item content validity index (I-CVI) and scale CVI (S-CVI) were performed. Expert analyses were found between 0.83 and 0.94 for I-CVI and 0.93 for S-CVI.

### 3.1. Exploratory Factor Analysis

Exploratory factor analysis was performed using principal components analysis to test the construct validity of the ESVADOPA scale in the passive phase. The KMO value was calculated as 0.89 as a result of the analysis, and Bartlett’s test of sphericity was found to be significant (*p* < 0.001). The results showed that the scale was composed of a one-factor structure that had an eigenvalue of over 1 and explained 71.79% of the variance. Cronbach’s alpha internal consistency coefficient of the scale is 0.92. An analysis of the factor structure showed that the item factor loads ranged between 0.729 and 0.897 (Table 2).

In the active phase, exploratory factor analysis was performed using the principal components analysis method to test the construct validity of the ESVADOPA scale. The KMO value was calculated as 0.86, and the Bartlett’s test of sphericity result was found to be significant (*p* < 0.00). The results showed that the scale was composed of a one-factor structure that had an eigenvalue of over 1 and explained 62.88% of the variance. Cronbach’s alpha internal consistency of the scale was found to be 0.87. When the factor structure was analyzed, the factor loads of the items were found to range between 0.666 and 0.879 (Table 3).

### 3.2. Convergent Validity

In both the active and passive phases, this study analyzed the correlation between the ESVADOPA scale items and VAS scores. Pearson correlation was used for analysis. The results indicated significant and positive relationships between all the variables. Positive correlations between all scale items and VAS can be considered to be evidence for the convergent validity of the scale (Table 4).

### 3.3. Confirmatory Factor Analysis

This study also performed confirmatory factor analysis using the maximum likelihood method for both passive and active phase measurements of the ESVADOPA scale. Errors in facial muscles and body response expressions were associated with each other during the confirmatory factor analysis performed for the passive phase. When the obtained results were analyzed, the data were found to have a good fit with the model: χ^2^ (8, N = 158) = 15.23, *p* > 0.05, (χ^2^/df = 1.90, GFI = 0.97, AGFI = 0.92, NFI = 98, TLI = 0.98, CFI = 0.99, and RMSEA = 0.08). As can be seen in Figure 1, factor loads of the scale range between 0.67 and 0.88 (Figure 1).

In the confirmatory factor analysis performed for the active phase, errors of facial muscles and body response expressions and errors of ability to relax and vegetative symptoms were associated with each other. Analysis results showed that the data had a good fit with the model: χ^2^ (7, N = 158) = 17.45, *p* < 0.05, χ^2^/df = 2.49, GFI = 0.97, AGFI = 0.91, NFI = 97, TLI = 0.96, CFI = 0.98, and RMSEA = 0.07. As seen in Figure 1, the factor loads of the statements ranged between 0.58 and 0.88 (Figure 2).

## 4. Discussion

This study adapted and tested the ESVADOPA scale for its Turkish version. The findings of this study confirmed that the Turkish version of the ESVADOPA scale was a reliable childbirth pain measurement tool. The findings support the one-factor structure of the scale.

The average intensity of childbirth pain is closely associated with the progression of labor and increases with more cervical dilation. Even if the pain increases in time, individual progression demonstrates differences from woman to woman according to both biological factors and the interventions performed [13]. Therefore, the dynamic and progressive nature of the childbirth pain is captured insufficiently, particularly by the visual/analog pain severity scales [14]. Analog pain severity scales do not consider interactions among factors such as contraction severity, presence of any relaxation between the contractions, the relative duration of this relaxation, total childbirth pain duration and coping, intrapartum support, and availability of the woman to respond [7]. Although the A-LPQ was the first scale to be developed specifically for labor pain, it similarly provides measurement based on the individual’s statement [9,10]. ESVADOPA, on the other hand, makes an assessment based on observation, does not have a language barrier, and is easily applied, which makes it a comprehensive alternative [1]. This study showed that the Turkish version of the ESVADOPA scale was reliable for midwives who played a key role in labor.

After the first translation of the scale to Turkish by the researchers, it was reviewed by a group of experts to enhance the content validity. The CVI value of the original scale was found to be 0.78. The I-CVI value of the Turkish version ranged between 0.83 and 0.94, and the S-CVI value was found to be 0.93. Researchers recommend that a scale with excellent content validity should have I-CVI values of 0.78 or higher and an S-CVI value of 0.9 and higher [15]. The findings indicate that the compatibility between the experts was high; the scale items were compatible with Turkish culture and represented the desired field; and the content validity was achieved.

Cronbach’s alpha internal consistency coefficient of the ESVADOPA was 0.784, which was over the cut-off point [1]. This result indicates a good level of reliability; namely, it indicates that the scale is a good measurement tool for the measurement of the expression of pain during childbirth [16]. This study found Cronbach’s alpha internal consistency coefficient of the Turkish version of the scale as 0.87, which was higher than that of the original scale. Cronbach’s alpha coefficients of between 0.60 and 0.80 indicate that the scale is highly reliable [17]. These findings indicate that the Turkish version of the scale was highly reliable.

EFA results indicated that the factor loads of the items in the scale were higher than 0.55. In both measurement times, the scale was found to have a single-factor structure. When the factor structure was analyzed, factor loads of the items were found to range between 0.729 and 0.897 (Cronbach α: 0.92) in the passive phase and between 0.666 and 0.879 (Cronbach α: 0.87) in the active phase. The KMO value was reported to be 0.796 in the original scale [1]. The KMO value was found to be 0.89 in the first measurement time and 0.83 in the second measurement time. Researchers accept values below 0.6 as moderate. Factor loads of over 0.60 are high, and they are considered medium levels if they are over 0.30. If the KMO value is close to 1, the data have a perfect fit with the factorial model [18,19]. These results indicate that the Turkish version of the scale had a strong factor structure.

When the convergent validity of ESVADOPA-TR was confirmed, like in the original scale, it analyzed its correlation with VAS, which is commonly used for assessing pain. The correlation coefficient was found r = 0.641 in the original scale, and it was found r = 0.635 in the Turkish version. The positive correlation between the two scales can be considered evidence for the convergent validity of the scale.

The scale measures the expression of pain in six dimensions and at four different levels. The literature determined the minimum value for the factor load of the items in the measurement tools between 0.30 and 0.40 in the literature [20]. According to Kline [19], factor loads of over 0.50 are adequate. The factor loads of the expressions in the study findings were found to range between 0.58 and 0.88. The literature indicates that the model compliance indicators should be >0.90 and RMSA < 0.08 [21,22,23]. The CFA results of this study showed that the data had good compliance with the model and explained its own factors adequately (χ^2^ = 17.45, *p* < 0.05, χ^2^/df = 2.49, GFI = 0.97, AGFI = 0.91, NFI = 97, TLI = 0.96, CFI = 0.98, RMSEA = 0.07). As no CFA was performed on the original scale, the results could not be compared with the original scale.

Test–retest is one of the best methods to show the invariance of the scales. Generally, a two-week duration between the first and second measurements is considered appropriate [17,24]. However, due to the nature of childbirth, it is not possible to have such a duration between the measurements. For this reason, repeated measures were performed to test the invariance of the scale in the active and passive phases. According to the test–retest analysis results in this study, there was a strong and positive relationship between the test–retest scores (*p* < 0.001). No test–retest was performed on the original scale. For this reason, the adaptation of the Turkish version is the first study that tested the time invariance of the scale.

## 5. Conclusions

The Turkish version of the ESVADOPA scale was found to have high reliability and validity for determining the expression of pain during childbirth. This tool is a valid and reliable tool that can be used by midwives to determine pregnant women’s pain levels during childbirth. The scale is ideal not only for the care provided by midwives but also for other professionals who provide assistance during labor. This includes anesthetists, as the scale improves the prediction for the duration of epidural anesthesia and helps to decrease potential dystocia with its rapidly applied feature. It is recommended to disseminate the use of this scale for the evaluation of specific and dynamic pain like childbirth pain and enable the experts from other specialization areas to evaluate the use of the scale.

## Figures and Tables

**Figure 1 healthcare-12-01745-f001:**
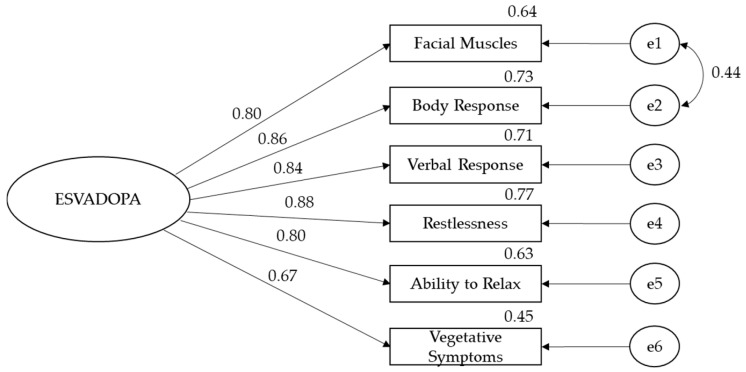
Confirmatory factor analyses of ESVADOPA in the passive phase; e1–e6: ESVADOPA item 1–6.

**Figure 2 healthcare-12-01745-f002:**
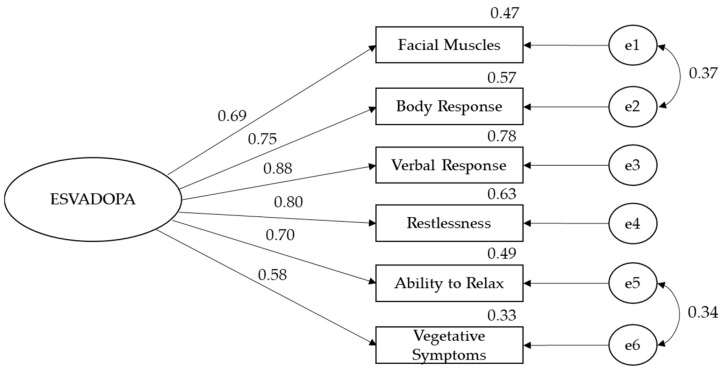
Confirmatory factor analyses of ESVADOPA in the active phase; e1–e6: ESVADOPA item 1–6.

**Table 1 healthcare-12-01745-t001:** Demographic characteristics of participants.

Variables	Mean (SD)/N (%)
**Age (year)**	26.87 (6.50)
**Gestational age (week)**	39.21 (0.8)
**Ethnicity** Turkish Syrian	98 (62%) 60 (38%)
**Education** Literate Primary Secondary High School University	17 (10.8%) 59 (37.3%) 38 (24.1%) 40 (25.3%) 4 (2.5%)

SD = standard deviation, N = number.

**Table 2 healthcare-12-01745-t002:** Descriptive statistics, internal consistency, and matrix of components of the exploratory factor analysis of ESVADOPA in the passive phase.

Items	Mean	SD	Corrected Item—Total Correlation	Cronbach’s Alpha If Item Deleted	Factor Loadings
Facial Muscles	1.26	0.733	0.783	0.905	0.857
Body Response	1.19	0.838	0.838	0.896	0.897
Verbal Response	0.92	0.859	0.796	0.903	0.864
Restlessness	1.13	0.838	0.828	0.898	0.888
Ability to Relax	1.01	0.740	0.764	0.907	0.837
Vegetative Symptoms	0.63	0.786	0.634	0.924	0.729

SD = standard deviation.

**Table 3 healthcare-12-01745-t003:** Descriptive statistics, internal consistency, and matrix of components of the exploratory factor analysis of ESVADOPA in the active phase.

Items	Mean	SD	Corrected Item—Total Correlation	Cronbach’s Alpha If Item Deleted	Factor Loadings
Facial Muscles	2.16	0.624	0.616	0.854	0.761
Body Response	2.18	0.613	0.688	0.843	0.813
Verbal Response	2.05	0.747	0.798	0.820	0.879
Restlessness	2.17	0.599	0.745	0.835	0.839
Ability to Relax	1.98	0.682	0.699	0.839	0.782
Vegetative Symptoms	1.49	0.943	0.553	0.880	0.666

**Table 4 healthcare-12-01745-t004:** Pearson correlation between the items of ESVADOPA and VAS in the passive and active phases.

Items	1	2	3	4	5	6	7	8	9	10	11	12	13
1-Facial Muscles-PP	1												
2-Body Response-PP	0.821 ***	1											
3-Verbal Response-PP	0.681 ***	0.703 ***	1										
4-Restlessness-PP	0.721 ***	0.780 ***	0.732 ***	1									
5-Ability to Relax-PP	0.628 ***	0.653 ***	0.703 ***	0.675 ***	1								
6-Vegetative Symptoms-PP	0.468 ***	0.572 ***	0.558 ***	0.579 ***	0.600 ***	1							
7-VAS-PP	0.635 ***	0.725 ***	0.725 ***	0.721 ***	0.707 ***	0.608 ***	1						
8-Facial Muscles-AP	0.509 ***	0.527 ***	0.500 ***	0.533 ***	0.437 ***	0.290 ***	0.489 ***	1					
9-Body Response-AP	0.521 ***	0.504 ***	0.475 ***	0.537 ***	0.416 ***	0.363 ***	0.525 ***	0.692 ***	1				
10-Verbal Response-AP	0.395 ***	0.483 ***	0.433 ***	0.529 ***	0.425 ***	0.315 ***	0.495 ***	0.612 ***	0.676 ***	1			
11-Restlessness-AP	0.464 ***	0.506 ***	0.460 ***	0.513 ***	0.426 ***	0.420 ***	0.511 ***	0.592 ***	0.610 ***	0.678 ***	1		
12-Ability to Relax-AP	0.367 ***	0.385 ***	0.421 ***	0.451 ***	0.518 ***	0.450 ***	0.511 ***	0.427 ***	0.511 ***	0.627 ***	0.569 ***	1	
13-Vegetative Symptoms-AP	0.231 ***	0.326 ***	0.341 ***	0.329 ***	0.402 ***	0.522 ***	0.425 ***	0.269 ***	0.335 ***	0.544 ***	0.506 ***	0.599 ***	1
14-VAS-AP	0.396 ***	0.428 ***	0.338 ***	0.474 ***	0.245 **	0.211 **	0.510 ***	0.635 ***	0.633 ***	0.525 ***	0.554 ***	0.351 ***	0.201 *

*** *p* < 0.001, ** *p* < 0.01, * *p* < 0.05. PP: passive phase. AP: active phase. VAS: Visual Analog Scale.

## Data Availability

Data will be available for other researchers upon request to the corresponding authors.

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
