# Peer review of "Psychometric Properties of the Turkish Version of the Rating Scale of Pain Expression during Childbirth Scale (ESVADOPA-TR)"

_healthcare, 2024, doi:10.3390/healthcare12171745_

Round 1

Reviewer 1 Report

Comments and Suggestions for Authors

I would like to express my sincere gratitude for giving me the opportunity to review this article. It was a privilege to be able to review this work and contribute to the advancement of research in this area.

The article is remarkably well structured and demonstrates a deep understanding of the topic addressed. The quality of the research performed and the clarity in the presentation of the results are truly impressive.

The way in which the scientific aspects were addressed reflects a high level of scientific rigor and dedication. I hope that my comments and suggestions will be useful and will contribute positively to the improvement of this work. I remain available for any additional clarifications or discussions that may arise during the review process.

Abstract:

Only the design was described, but not the type of study.

Materials and Methods:

1- Describe the type of study. I understand that this is a methodological study.

2- Is it possible that there is bias in filling out the scale, since it is filled out by a third party and not by the patient herself? Since pain is subjective and can only be felt by the person herself?

3- Two measurements, such as passive and active phases, were performed for test-retest. The maximum likelihood test was used for test-retest reliability measurements. How was temporal stability achieved, considering that the ideal interval according to the literature is between 7-14 days?

4- Describe the passive and active phases in more detail.

Results

1- The statement in Table 1 is incorrect.

2- I understand that the ESVADOPA scale is multidimensional. Has this been confirmed?

3- It is stated that the Cronbach's alpha of the scale is 0.92, but in Table 2 this value corresponds to Vegetative Symptoms. In this case, I understand that the highest Cronbach's value was described, but I would like to know if the total value was calculated, since only the values ​​of each dimension were described in Table 2. I suggest including the total Cronbach's value in the table.

4- The Shapiro-Wilks test was performed to determine the normality distribution. However, this data was not mentioned in the results. It is assumed that the data follow a normal distribution, since Pearson's Correlation was used. Could you confirm this data, please?

5- Describe the type of correlation that was used in the title of Table 4.

References

I suggest trying to use more up-to-date references. More than half of the references cited are more than five years old.

Author Response

Comments 1: Only the design was described, but not the type of study.

Response 1: We have accordingly revised as “This study utilized a methodological design.” Page 1, line 18.

Comments 2: Describe the type of study. I understand that this is a methodological study.

Response 2: Agree. We have accordingly changed as “This study utilized a methodological design…..”  Page 2, line 87.

Comments 3: Is it possible that there is bias in filling out the scale, since it is filled out by a third party and not by the patient herself? Since pain is subjective and can only be felt by the person herself?

Response 3 : We agree with you. Pain is subjective. However, labor pain is different from other pains. Its intensity varies from the beginning to the end of the labor process. A pregnant woman who experiences this pain for the first time may describe the intensity of the pain as a 10 even with 1-2 cm dilatation. However, as the process progresses, she may describe the increasing pain as a 10 again. This situation can be misleading in evaluating the pain of the pregnant woman. Therefore, ESVADOPA is unique and functional in that it allows health professionals to evaluate pain expressions.

Comments 4: Two measurements, such as passive and active phases, were performed for test-retest. The maximum likelihood test was used for test-retest reliability measurements. How was temporal stability achieved, considering that the ideal interval according to the literature is between 7-14 days?

Response 4: The specified period is recommended to re-test the invariance over time (7-14 day). However, it is not possible to use the specified period due to the nature of labor. Only hours pass between the passive and active phases. It is a questionnaire that measures labour pain, it must be taken into account the progression of labour. The active and passive phase is the 2nd stage of labor.   Therefore, we measured in two stages of labor.

Comments 5: Describe the passive and active phases in more detail.

Response 5: Agree. It was revised “Data collection was carried out during the first stage of labor.  To test the invariance of the scale over time, two measurements were performed at passive and active labor phases. The passive phase was the time from the beginning of labor to 5 cm cervical dilatation, while the active phase began at 6 cm cervical dilatation to birth.”   Page3, Line 134-137.

Comments 6: The statement in Table 1 is incorrect.

Response 6: Agree. I was corrected. Page 6, line 196.

Comments 7: I understand that the ESVADOPA scale is multidimensional. Has this been confirmed?

Response 7: In fact, the scale is single-factor. It hasn't got any sub-dimension. Scale items have names other than numbers. Each item evaluates a pain expression in the face, body, verbal expressions, restlessness, relaxation, and vegetative symptoms. Because of their names, they may be understood as sub-dimensions.

Comments 8: It is stated that the Cronbach's alpha of the scale is 0.92, but in Table 2 this value corresponds to Vegetative Symptoms. In this case, I understand that the highest Cronbach's value was described, but I would like to know if the total value was calculated, since only the values ​​of each dimension were described in Table 2. I suggest including the total Cronbach's value in the table.

Response 8: We checked the data results again. It is correct. The total Cronbach α value is 0.92. We have declared that we will present our data set for checking if needed.

Comments 9: The Shapiro-Wilks test was performed to determine the normality distribution. However, this data was not mentioned in the results. It is assumed that the data follow a normal distribution, since Pearson's Correlation was used. Could you confirm this data, please?

Response 9: It was added in method> data analysis. page 4 line 170-172.

Comments 10: Describe the type of correlation that was used in the title of Table 4.

Response 10:  It was added. Page 6- line 222,226.

Comments 11:  I suggest trying to use more up-to-date references. More than half of the references cited are more than five years old.

Response 11: References were checked. Most of the references that are more than 5 years old belong to statistics and standard definitions. If we prefer a current source, we will not cite the first author.

Reviewer 2 Report

Comments and Suggestions for Authors

A STROBE checklist is needed to this cross-sectional study.

Your sample size calculation should be based on the instrument ICC and the number of items to be more specific on measuring the reliability of ESVADOPA.

Having only one rater (midwife in this case) can introduce the possibility of rater bias, which is not being considered in this study. The perception and assessment of pain is rather subjective, and different raters may assess it differently, and it is also more practical in clinical settings whereby different healthcare professionals will be in charge of the pain assessment during different time points.

In terms of the criterion validity, I notice that only VAS was used and measured it correlation with ESVADOPA. How about other labor pain measures which were mentioned in your introduction, e.g., MPQ, A-LPQ?

Author Response

Comments 1:  A STROBE checklist is needed to this cross-sectional study.

Response 1: STROBE checklist added.

Comments 2: Your sample size calculation should be based on the instrument ICC and the number of items to be more specific on measuring the reliability of ESVADOPA.

Response 2: The sample size explanation was revised. Calculation with ICC has been added and the reference has been updated. Page 3, line100-103, 373.

Comments 3: Having only one rater (midwife in this case) can introduce the possibility of rater bias, which is not being considered in this study. The perception and assessment of pain is rather subjective, and different raters may assess it differently, and it is also more practical in clinical settings whereby different healthcare professionals will be in charge of the pain assessment during different time points.

Response 3: We agree with this comment. At the beginning of the study, we considered having different raters such as midwives, nurses, and doctors. However, we decided that this could cause bias. As you stated, pain is already subjective. And as you said, different raters may assess it differently. In this case, confounding factors will come into play (such as experience, age, professional differences, and level of education). Since this was considered risky in terms of the reliability of the study, we conducted it with a single rater.

Comments 4: In terms of the criterion validity, I notice that only VAS was used and measured it correlation with ESVADOPA. How about other labor pain measures which were mentioned in your introduction, e.g., MPQ, A-LPQ?

Response 4: We believe that the MPQ is not a suitable scale for assessing labor pain. Although the MPQ is quite comprehensive, its pain definitions are very diverse. Its items are pain expressions that are not usually used to describe labor pain. There is no Turkish language validity study for the A-LPQ scale yet. If there was, this scale would be our choice. Perhaps we can evaluate both in the future.

Round 2

Reviewer 2 Report

Comments and Suggestions for Authors

N.A.

Comments on the Quality of English Language

N.A.